# Factor V Leiden, Factor II, Protein C, Protein S, and Antithrombin and Ischemic Strokes in Young Adults: A Meta-Analysis

**DOI:** 10.3390/genes13112081

**Published:** 2022-11-09

**Authors:** Mihael Tsalta-Mladenov, Mariya Levkova, Silva Andonova

**Affiliations:** 1Department of Neurology and Neuroscience, Medical University “Prof. Paraskev Stoyanov”, Marin Drinov St. 55, 9000 Varna, Bulgaria; 2Second Clinic of Neurology with ICU and Stroke Unit, University Multiprofile Hospital for Active Treatment “St. Marina”, Hristo Smirnenski Blvd. 1, 9000 Varna, Bulgaria; 3Department of Medical Genetics, Medical University “Prof. Paraskev Stoyanov”, Marin Drinov St. 55, 9000 Varna, Bulgaria; 4Laboratory of Medical Genetics, University Multiprofile Hospital for Active Treatment “St. Marina”, Hristo Smirnenski Blvd. 1, 9000 Varna, Bulgaria

**Keywords:** stroke, young, thrombophilia, factor V Leiden, prothrombin, protein C, protein S, antithrombin

## Abstract

Ischemic strokes are one of the leading causes of death worldwide. The aim of this meta-analysis is to elaborate on the role of inherited predisposition to thrombophilia in the etiology of ischemic strokes in young adults. The keywords factor V Leiden (FVL), factor II, prothrombin (PT), protein C (PC), protein S (PS), antithrombin (AT), ischemic stroke, and young were used to search different databases. We selected studies with participants who were between 18 and 65 years. A total of 104 studies were eligible for inclusion in the meta-analysis. All the studied genetic markers were risk factors for ischemic stroke according to our results (FVL OR = 1.74; PT OR = 1.95; PC OR = 10.20; PS OR = 1.74; AT OR = 3.47; *p* < 0.05). There was moderate heterogeneity for most of the results, and subgroup analyses were conducted by dividing the studies according to the geographic location, gender ratio, and selection criteria of the performed study. There were no significant differences between the groups, but different geographic location was a probable source of heterogeneity. All of the studied markers—FVL, prothrombin, PC, PS, and AT—were significantly associated with increased risk of ischemic stroke in young adults and, if tested, could improve the quality of care.

## 1. Introduction

Ischemic strokes are one of the leading causes of death and reduced quality of life worldwide. Although they are traditionally associated with advanced age, more than two million young adults are diagnosed with an ischemic stroke (IS) each year [1]. In Bulgaria, the incidence of ischemic strokes among people under the age of 55 was around 4000 people in 2021, accounting for roughly 10% of all IS patients [2].

When ischemic strokes affect the younger population, they have more severe economic consequences because the affected individuals may not fully recover, may be unable to work, and require someone to help them in their daily activities [3].

Because of this, investigation of the etiology of ischemic strokes among young people is of the utmost importance. Risk factors could be traditional ones, such as hypertension, hyperlipidemia, and obesity, but may also include persistent foramen ovale, dissection of the aorta, drug abuse, pregnancy, etc. [1]. Nevertheless, in around one-third of all cases of ischemic strokes in young adults, the risk factors are unknown [4].

In the past few years, a lot of attention has been paid to genetic predisposition to thrombophilia—for example factor V Leiden, prothrombin gene mutations, and deficiencies in protein C, protein S, antithrombin, etc. [5].

Factor V Leiden (FVL) is a genetic variant of normal factor V. As a result of amino acid substitution, factor V cannot be inactivated by activated protein C (APC), which leads to APC resistance and a genetic predisposition to thrombophilia. People who are heterozygous carriers of FVL are four times more likely to present with venous thromboembolism [6].

Variants in the prothrombin gene are another reason for a genetic predisposition to thrombophilia. These mutations result in higher plasma levels of prothrombin due to an elevated expression of messenger RNA [6]. Prothrombin can turn into thrombin, thus increasing the risk of thrombosis by approximately 2–4 times when the variant is in a heterozygous state [6].

Deficiencies in proteins C and S, as well as in antithrombin, show a 50-times higher chance of thrombosis compared to FVL and prothrombin [6]. In addition, in most cases, these deficiencies are reported based only on laboratory findings, and genetic analyses are performed very rarely [6].

Both protein C and S are produced by the liver, and this process depends on vitamin K. PC is first activated into APC, which acts by inactivating factors Va and VIIIa, thus preventing thrombosis. Protein S (PS) is a cofactor of PC [7]. Depending on the type of mutation of the genes regulating PC, its deficiency can be a mild one, with a prevalence of 1 in 500 individuals. The severe type is relatively rare—it affects one in every four million people. A mild deficiency in PS is observed in 1 in 500 people [8].

Antithrombin (AT) is mainly involved in the inactivation of thrombin and factor Xa. When AT binds to heparin, its anticoagulant effect is boosted [9]. AT deficiency can be either congenital due to different mutations or acquired when, for example, the production of AT in the liver is decreased or in cases of enormous protein loss [10]. The lack of AT predisposes to thrombophilia and is observed in around 1 in 2000 to 1 in 5000 individuals [8].

The aim of this paper is to determine how genetic predisposition to thrombophilia—specifically, the heterozygous or homozygous states of FVL and prothrombin, as well as levels of PC, PS, and AT—affects the etiology of ischemic strokes in young individuals.

## 2. Materials and Methods

The search strategy was peer-reviewed and registered with the International Prospective Register of Systematic Reviews. We performed a search of the MEDLINE (PubMed) and Web of Science databases in accordance with PRISMA guidelines [11] and used the following keywords, both alone and in combination with each other: factor V Leiden, factor II, prothrombin, protein C, protein S, antithrombin, ischemic stroke, and young. No restrictions on publication dates were applied. Only full-text articles published in English and reporting findings about correlations between FVL, prothrombin, PC, PS, or AT and ischemic strokes in young adults were included in the next steps. Review articles were excluded. The studies were first screened based on their titles and abstracts using Rayyan software [12]. Then, the full texts of the selected articles were further reviewed.

The following criteria for inclusion in the meta-analysis were used: 1. case–control studies regarding young adults with ischemic strokes who were tested for FVL, prothrombin, PC, PS, or AT were included, while cohort studies were excluded; 2. the indexed patients must be between the ages of 18 and 65; 3. animal-only studies were excluded; 4. reviews and meta-analyses were excluded; 5. the full information about the ages of the case subjects and the types of mutations must be provided by the authors; 6. the articles must be written in English, and the full text must be available.

We also collected information regarding the year of publication, the ethnic background, and the total number of the participants, as well as the age, gender, geographical setting, and health status of the included individuals. Studies that did not meet the inclusion criteria were excluded.

After selecting eligible studies, we performed a meta-analysis using Review Manager (RevMan) software (version 5.0. Copenhagen: The Nordic Cochrane Centre, The Cochrane Collaboration, 2008) and a fixed model. We found the values of odds ratio (OR), Z-test score, and heterogeneity for FVL, prothrombin, PC, PS, and AT and performed subgroup analyses in cases of high levels of heterogeneity. Our goal was to check the risk of stroke among young individuals who had genetic predispositions to thrombophilia due to changes in the above-mentioned markers compared to healthy control subjects. We applied a two-tailed *p*-value ≤ 0.05 and analyzed the included articles for publication bias using funnel plots. Using the genetic data provided in the publications, we conducted the meta-analysis. If the information was given as a prevalence of hetero- and homozygotes, we used allele frequency. We made the assumption that the gender ratio was equal if there was no information available regarding the precise number of men and women who took part in a study.

The quality of the included publications was evaluated using the Newcastle–Ottawa scale (NOS) [13]. The maximum rating for a study was nine stars. Two authors separately reviewed the quality assessment results. A study was considered of high quality when it received six or more stars.

## 3. Results

After the search was completed, there were 1294 results for FVL, 2043 results for prothrombin, 2975 results for both PC and PS, and 909 results for AT (Figure 1). After duplicates were excluded, we assessed the titles and abstracts of the records and rejected articles that did not meet the inclusion criteria. At the end, there were 104 eligible studies in total—42 for FVL [14,15,16,17,18,19,20,21,22,23,24,25,26,27,28,29,30,31,32,33,34,35,36,37,38,39,40,41,42,43,44,45,46,47,48,49,50,51,52,53,54,55], 30 for prothrombin [15,17,18,19,21,22,23,24,27,28,31,32,33,34,35,38,41,42,43,44,45,46,49,50,51,52,54,56,57,58], 10 for PC [19,20,21,28,37,39,47,53,59,60], 12 for PS [19,20,21,28,37,39,47,53,59,60,61], and 10 for AT [21,28,37,39,47,53,59,60,61,62,63]. The studies were focused on the following polymorphic variants: FVL G1691A and prothrombin G20210A. Deficiencies in PC were due to mutations in the *PROC* gene, those in PS were due to mutations in the *PROS1* gene, and those in AT were due to mutations in the *SERPINC1* gene. However, most studies about these three markers focused primarily on measuring their levels. The eligible studies were later used in the meta-analysis. The NOS was applied in order to check the quality of all the qualifying studies, and all of the publications received seven or more stars (Appendix A). In most of the studies, the included patients did not have any cardiovascular or cerebrovascular risk factors.

Due to the fact that not all of the articles for FVL and prothrombin reported their results divided into heterozygous and homozygous genotypes, we performed two meta-analyses for each of these factors. The first one included only the studies that grouped their subjects according to different genotypes for the studied marker, while the second one included only articles that classified their participants as mutant or healthy individuals.

According to the results, FVL increased the risk of ischemic stroke when both hetero- and homozygotes were reported (OR = 1.75, *p* < 0.05, Figure 2). The percentage of heterogeneity was 17%. We also performed a subgroup analysis by dividing the included articles based on the geographical location of the study—Europe and Asia, South, and North America (Figure 2). There were studies from Africa only for FVL and prothrombin [36,52]. There were no studies performed in Australia. There was no statistical difference between the two groups (*p* = 0.16), and the level of heterogeneity remained similar at 20% for the Europe group and 9% for the other group. We also separated the studies according to the gender ratio of the participants: one group for studies with equal gender ratios and another group for studies with unequal ratios, where men or women made up more than 60% of the cohort. However, there was no statistical difference between the two groups (*p* = 0.71), and the heterogeneity was also moderate at 24% for both groups (Appendix A). Then, we performed a subgroup analysis by dividing the studies into those with cryptogenic strokes and those with other risk factors, as well. There was not a statistical difference (*p* = 0.24); the heterogeneity for the first subgroup was moderate at 26%, while that of the other group was low at 17% (Appendix A).

After performing a meta-analysis on the articles reporting their participants as either mutant and healthy individuals only for FVL, there was a borderline statistically significant result: OR = 1.30, *p* = 0.05. The percentage of heterogeneity was 9% (Appendix A).

No matter whether the results were reported as homo- and heterozygotes or as mutated and healthy individuals, prothrombin was also found to be a risk factor for ischemic stroke. In the first case, the OR was 1.95, *p* < 0.05 (Figure 3), and in the second one, the OR = 5.16, *p* < 0.05 (Appendix A). The level of heterogeneity was 33% when the results included exact information about the genotypes of the participants and 85% when no such data were available. We again performed three different subgroup analyses by dividing the studies. The first was based on the geographic localization of the studies; the second was based on the gender ratio of the participants; and the third one was based on whether the studies recruited only patients with cryptogenic stroke (Appendix A). In the second case, there was not a statistical difference (*p* = 0.18), but the heterogeneity for the subgroup with equal gender ratios was low (2%), while it was high for the other group (61%). For the other subgroup analyses, there were no statistical differences between the two subgroups (*p* > 0.05). However, there were significant changes in heterogeneity: 11% for the “cryptogenic stroke” group, 43% for the “non-cryptogenic stroke“ group, 16% for the “Europe” group, and 65% for the “Asia, South, and North America” group.

Because of the high percentage of heterogeneity when participants were reported as mutant and healthy individuals, we performed a subgroup analysis and again divided the studies based on the geographical location of the study into a Europe group and an Asia, South, and North America group. However, the percentages of heterogeneity remained high: 88% and 81%, respectively (Appendix A). Because there was only one study in one subgroup and an analysis would not produce logical conclusions, we did not split the studies based on gender distribution or by those exclusively recruiting patients with cryptogenic strokes.

Articles on the correlation between strokes and PC, PS, and AT involved laboratory diagnostics without any genetic testing, and the results were divided into subjects with deficiencies in these markers and normal individuals. All three—PC, PS, and AT—were risk factors for an ischemic stroke (OR for PC = 10.20, OR for PS = 1.74, OR for AT = 3.47; *p* < 0.05; Figure 4, Figure 5 and Figure 6). The percentage of heterogeneity was low for PC at I^2^ = 38.00%, but for PS and AT, it was higher at 73.00% and 75.00%, respectively. We performed subgroup analyses for all three markers. We subdivided the studies into two groups based on the geographical location of the included participants—those from Europe and those from Asia, South, and North America. The results for PC showed a statistically significant difference between the two groups (*p* = 0.02) and a notable decrease in the levels of heterogeneity: 0% for the Europe group and 11% for the other group. As for PS, after this analysis, there was not a statistically significant difference between the two groups (*p* = 0.24), but there were differences in the levels of heterogeneity. For the Europe subgroup, I^2^ = 0%, and for the other group, I^2^ = 88%. As for AT, the two groups—Europe and Asia, South, and North America—also did not show a statistically significant difference (*p* = 0.81), and the percentages of heterogeneity remained high: I^2^ = 73% and I^2^ = 90%, respectively. This was why we divided the articles for AT according to the gender ratios of the participants: one group with equal gender ratios and the other with unequal ratios, where men or women made up more than 60% of the cohort. Despite showing no statistical differences between the two groups (*p* = 0.37), the heterogeneity of the first group was 0%, and of the second one was 84% (Appendix A).

The meta-analysis we performed showed that all of the studied genetic polymorphisms—FVL, prothrombin, PC, PS, and AT—were risk factors for ischemic stroke. The funnel plots were symmetrical, and there was no obvious risk of publication bias (Appendix A).

## 4. Discussion

There are no well-defined recommendations on genetic counseling and testing for thrombophilia in stroke patients. In the latest recommendations, genetic factors for ischemic strokes are only discussed as possible nonmodifiable risk factors for ischemic strokes [64,65,66]. The guidelines from the American Heart Association and the American Stroke Association for the prevention of strokes in patients with ischemic strokes or transient ischemic attacks (TIAs) from 2021 do not specify who should be tested for thrombophilia or what tests should be carried out. It is only stated that, in certain clinical scenarios mostly related to paradoxical emboli or venous thrombosis, testing for thrombophilic predisposition may be considered, as the test should be deferred or repeated at least 4 to 6 weeks after the acute phase because the results may change during this time [67]. This is why it is important to conduct a research study involving a large cohort of patients and control individuals in order to elaborate on the potential role of inherited thrombophilia in the clinical practice.

The results from our meta-analysis showed that FVL, prothrombin, PC, PS, and AT were all risk factors for ischemic strokes among young adults. This finding is in accordance with other reports. Chiasakul et al. also conducted a meta-analysis regarding the role of inherited thrombophilia in patients with arterial ischemic strokes and concluded that FVL, prothrombin, PC, PS, and AT correlated with an increased risk of stroke, especially in young patients [68]. Kim et al. reported that FVL and prothrombin were associated with an increased risk of ischemic stroke, mostly in people younger than 55 years old and in women [69]. Another research group confirmed the finding that prothrombin was a potential risk factor for ischemic stroke [57].

Another meta-analysis by Hamedani et al. focused specifically on the association between FVL and ischemic strokes in patients who were 50 years or younger. According to their results, FVL was a risk factor for ischemic stroke in the studied cohort, but this association was significant only for patients who were selected based on a clinical hypothesis of prothrombotic condition due to the coexistence of, for example, deep venous thrombosis. In patients without such history, it was less likely to find a correlation between FVL and ischemic strokes [70]. In our meta-analysis, such a subgroup analysis was not possible due to a lack of data, which was a limitation of the present work.

As for PC, PS, and AT, we found only one published meta-analysis analyzing the possible correlations between these factors and arterial ischemic strokes in young adults. The authors found that, in patients who were 65 years or younger, PC and PS deficiencies were possible risk factors for ischemic stroke: the ORs were 2.13 and 2.26, respectively [68]. We also reached the same conclusion. However, our OR for PC was much higher at 10.20, while for PS it was similar at 1.74. The much higher OR could be explained by the fact that we included more studies in our meta-analysis compared to the work by Chiasakul et al. Their analysis for PC was based on eight studies, while ours was based on ten [68].

According to Chiasakul et al., deficiency in AT did not show a statistically significant result [68]. On the contrary, we concluded that AT correlated with an increased risk of ischemic stroke, and the OR was equal to 3.47. This could be a result of the larger number of included studies in our analysis of ten, while the other research group based their analysis on only six studies.

Nevertheless, there are also reports that deny the roles of the studied factors for predisposition to thrombophilia. For example, Pahus et al. reported that heterozygote carriers of FVL were predisposed only to transient ischemic attacks [39]. There are also some hypotheses that FVL is associated with ischemic strokes, but only in the presence of patent foramen ovale [70].

It should be pointed out that there are also other risk factors that, in combination with a genetic predisposition to thrombophilia, may significantly increase the chance of an ischemic stroke. One such factor is, for example, PFO [70]. There were three articles in our meta-analysis that included patients with this condition [18,28,38]. However, we could not guarantee that the subjects in other case–control studies did not have PFO because its diagnosis requires additional testing, which may not have been performed. There are data about a possible correlation between PFO, FVL, and ischemic strokes [70]. This is why one could hypothesize that FVL alone is not enough to predispose an individual toward ischemic stroke and that it must be in combination with other risk factors. This is the reason why more case–control studies recruiting patients with cryptogenic strokes are needed in order to elaborate on the roles of combined risk factors. Another risk factor is hyperhomocysteinemia, which may be due to a C677T variant in the *MTHFR* gene. However, according to a large meta-analysis focused on venous thromboembolisms, this variant did not predispose patients toward thrombophilia, even in combination with changes in the genes for FVL or prothrombin [71]. This is why we did not include this polymorphism in our meta-analysis. Moreover, there are new candidate genes identified by Genome-Wide Association Studies (GWAS), such as factors XI and XII [72].

One should also keep in mind that testing for inherited thrombophilia can be very expensive and is not covered by health insurance in all countries. Furthermore, even if a patient tests positive for thrombophilia, the treatment plan is rarely affected [73]. It was estimated that only one out of twelve young patients who presented with an ischemic stroke received different treatment because of a positive test for inherited thrombophilia [74]. Despite all of the above, testing for inherited thrombophilia is performed in clinical practice, especially in young stroke patients. This is why clinicians should be aware of its utility, making the results from the current meta-analysis relevant to everyday practice.

Our research had several limitations. We reported moderate percentages of heterogeneity after the first step of the meta-analysis was completed. During the subgroup analyses, we divided the studies based on the geographic location of the research, the gender ratio of the participants, and the selection criteria of the performed study. There were no statistical differences between most of the analyzed subgroups. However, the levels of heterogeneity decreased. This illustrated that a different study design may have a significant impact on the results. Despite the fact that the meta-analysis was based on similar case–control studies, the selection criteria of the research groups might have slight differences, for example, because different control groups were used, the participants had different ethnic backgrounds, etc. Moreover, there are data suggesting that the prevalence of genetic predisposition to thrombophilia may differ based on geographic localization. For example, FVL is more prevalent among people of European descent and is rarely reported in patients of sub-Saharan African and east Asian descent. Even in Europe, its distribution is not equal, and there is a possible positive gradient of prevalence from south to north [75]. On the contrary, protein C and S deficiencies are reported to be rare in Caucasians but are more prevalent in Japan, Taiwan, and Thailand [74]. Our meta-analysis combined studies from different countries. If the allele associated with the disease is more common in a certain population compared to another, there is a higher chance of finding an association in that particular population. Moreover, Caucasians suffer more often from thromboembolisms compared to other ethnic groups [75]. Even though we did not find a statistical difference between the subgroups based on geographic localization, with the exception of PC, all these factors could serve as potential sources of heterogeneity.

Another important source of heterogeneity was the risk profile of the included participants. In most of the case–control studies, classic risk factors for stroke, such as hypertension, diabetes, hyperlipidemia, etc., were more prevalent among patients with ischemic strokes. These could have an impact on the incidence of stroke and increase stroke risk, as well as the genetic predisposition to thrombophilia. This was also noted by Hamedani et al., who published a meta-analysis on inherited thrombophilia among young stroke patients. They reported a significant association between FVL and the risk of stroke with an odds ratio was 1.4. However, because of high heterogeneity among the included studies, the authors performed subgroup analyses, selecting studies with cryptogenic strokes. The odds ratio for FVL among patients with cryptogenic strokes was higher than that in the first case at 2.7 [70]. This is why more case–control studies involving only participants with cryptogenic strokes are needed in order to better estimate the role of genetic predisposition to thrombophilia.

Another limitation of our work was that, although we found a large number of studies for FVL and prothrombin, some of them reported their results as mutant or healthy individuals only and did not contain information about the exact genotypes of the participants. This is why we could not perform a meta-analysis with all of the studies but had to separate them in two different groups based on the presence or absence of full genomic data of the included patients. The significance of the current meta-analysis could have been higher if we used a larger number of studies. This is why it is extremely important to follow universal guidelines and to publish full genomic data in order to increase the transparency of results. In addition, we did not perform subgroup analyses on age or the presence of PFO because exact information was missing from the studied articles. It would be interesting to see how age and PFO affect the association between thrombophilia and ischemic strokes.

In addition, it should be pointed out that most of the case–control studies on the roles of PC, PS, and AT did not perform genetic testing but instead analyzed the plasma activity or concentrations of these markers and reported deficiencies in the event of reductions in the above-mentioned parameters.

Regardless of its limitations, our meta-analysis summarized the largest number of case–control studies on genetic predisposition to thrombophilia among young stroke patients to date. We report significant results about the roles of FVL, prothrombin, PC, PS, and AT as possible risk factors for ischemic stroke. Because the included case–control studies were conducted in many different countries, the results could be applied to clinical practice, regardless of location.

## 5. Conclusions

The number of young stroke patients is increasing, thus requiring more investigation of the possible risk factors. Inherited thrombophilia is one of them, and it may be associated with ischemic strokes due to hypercoagulability state. According to our results, FVL, prothrombin, PC, PS, and AT could all predispose toward ischemic strokes among young adults and should be part of laboratory testing in cases of young stroke patients. More explicit recruiting criteria, as well as a universal way of reporting genetic results, in case–control studies are needed in order to better evaluate the roles of these thrombophilic factors in the etiology of ischemic strokes among young patients.

## Figures and Tables

**Figure 1 genes-13-02081-f001:**
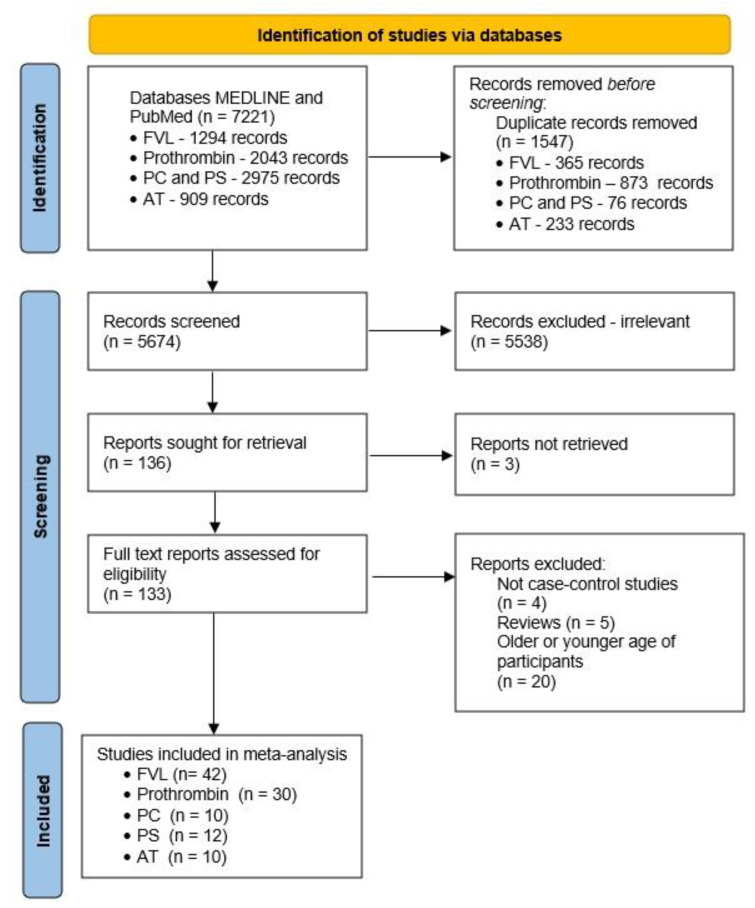
Flow diagram of the process of selecting eligible articles for inclusion in the study.

**Figure 2 genes-13-02081-f002:**
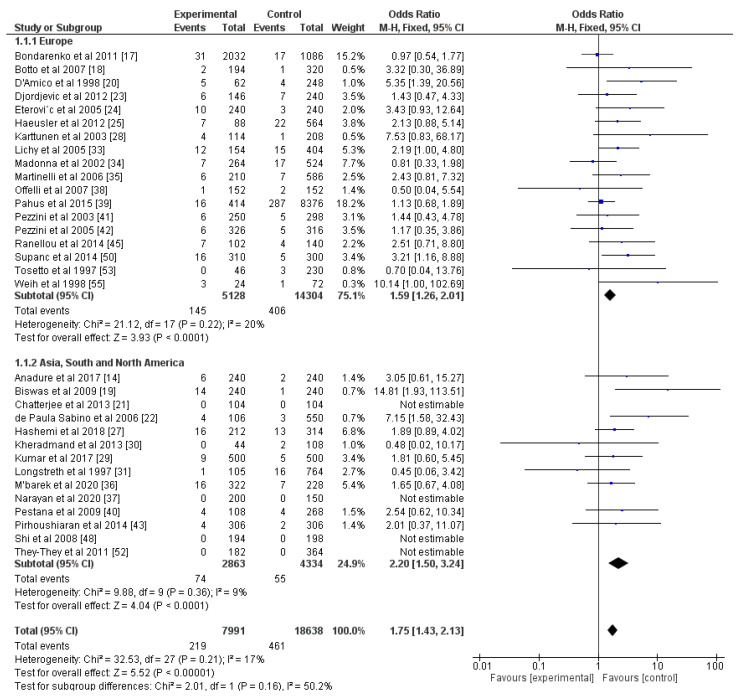
Forest plot for FVL with studies reporting prevalence of hetero- and homozygotes, and subgroup analysis according to geographical localization of studies. All included articles are presented with the corresponding numbers of mutated alleles in the experimental group and the control group, as well as the total number of alleles. Blue squares indicate odds ratios. Black rhombi indicate overall effect. Vertical line indicates null effect, and horizontal lines indicate values of odds ratios.

**Figure 3 genes-13-02081-f003:**
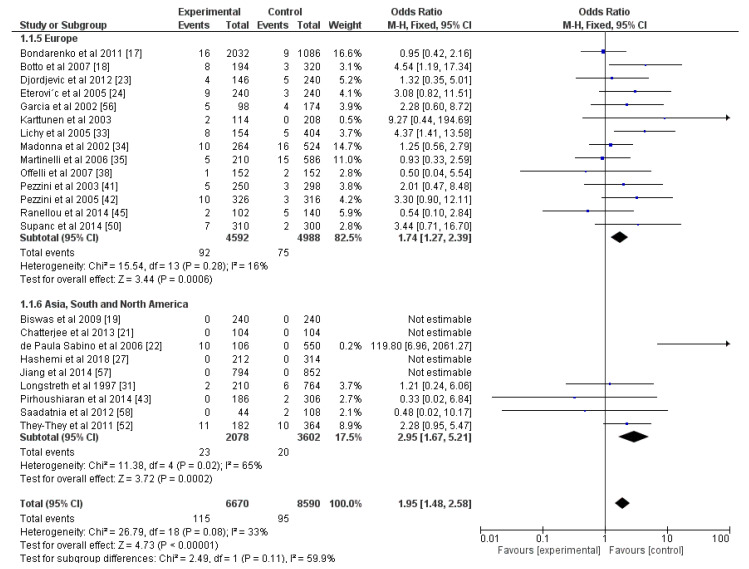
Forest plot for prothrombin with studies reporting prevalence of hetero- and homozygotes, and subgroup analysis for geographic localization of the studies. All included articles are presented with the corresponding numbers of mutated alleles in the experimental group and the control group, as well as the total number of alleles. Blue squares indicate odds ratios. Black rhombi indicate overall effect. Vertical line indicates null effect, and horizontal lines indicate values of the odds ratios.

**Figure 4 genes-13-02081-f004:**
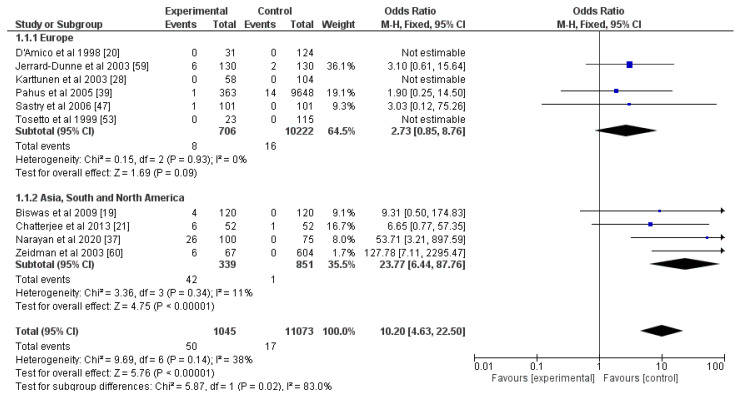
Forest plot for protein C and subgroup analysis based on geographic location of the study. Blue squares indicate odds ratios. Black rhombi indicate overall effect. Vertical line indicates null effect, and horizontal lines indicate values of odds ratios.

**Figure 5 genes-13-02081-f005:**
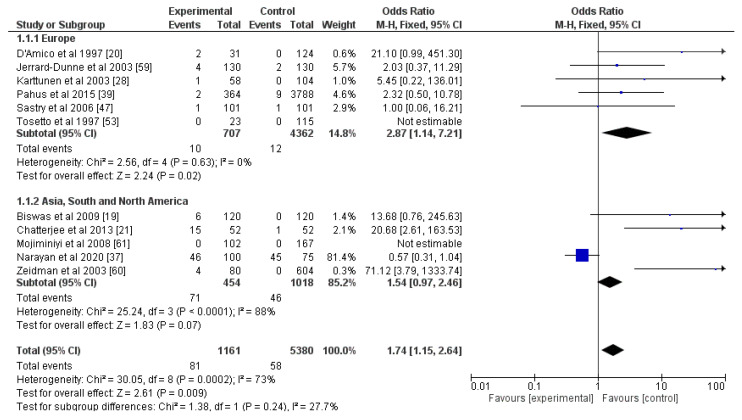
Forest plot for protein S and subgroup analysis based on geographic location of the study. Blue squares indicate odds ratios. Black rhombi indicate overall effect. Vertical line indicates null effect, and horizontal lines indicate values of odds ratios.

**Figure 6 genes-13-02081-f006:**
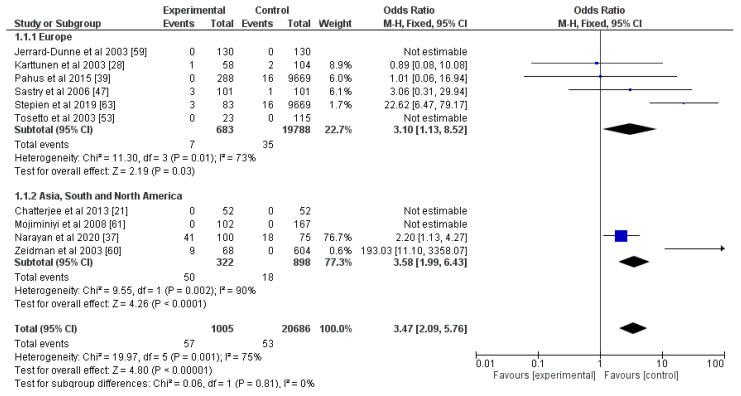
Forest plot for antithrombin and subgroup analysis according to geographical localization of the studies. Blue squares indicate odds ratios. Black rhombi indicate overall effect. Vertical line indicates null effect, and horizontal lines indicate values of odds ratios.

## Data Availability

Not applicable.

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
