# Peer review of "Factor V Leiden, Factor II, Protein C, Protein S, and Antithrombin and Ischemic Strokes in Young Adults: A Meta-Analysis"

_genes, 2022, doi:10.3390/genes13112081_

Round 1

Reviewer 1 Report

 See the comments below.

In this large meta-analysis conducted in a population affected by cryptogenic stroke, the role of some markers associated with haemocoagulative disorders was taken into consideration, confirming that the incidence of cryptogenic events is a rather frequent clinical occurrence and even more so, with a high clinical impact if we think about it. which often concerns young subjects who are not only at risk of a first ischemic episode but also at risk of stroke recurrence. However, the authors should take into account that patients with cryptogenic events often have the lowest prevalence of risk factors for atherosclerosis and the lowest frequency of atherosclerotic disease. Furthermore, there are often patients where there is no comorbidity and have a low risk of acute coronary events compared to patients with events of known cause. Similarly, patients with cryptogenic events had a low long-term risk of new atrial fibrillation (AF), a low risk of paroxysmal AF, or lower-risk cardiac abnormalities at baseline. Most of the previous studies addressing risk factors on the prognosis of cryptogenic stroke were based on research conducted in hospitals, without separating truly cryptogenic cases with detailed diagnostic work from cases with more than one cause or cases with unknown causes due to incomplete investigations. Therefore, the authors should provide clarification on the risk classes of the populations under study, or at least, if this is not possible, they should emphasize the fact that these patients are often naive in terms of cardiovascular and cerebrovascular risk. From a methodological point of view, there is excellent scientific rigor in the selection and analysis of the data from each study and the subdivision into geographical areas is also very relevant. The study, therefore, offers interesting insights that pose a serious reflection on the fact that screening of these factors could help prevent a greater number of cerebrovascular events of unknown etiology.

Author Response

Dear reviewer, thank you for your comment. In most of the studies the included patients did not have any cardiovascular and cerebrovascular risk factors. Unfortunately, in some of the articles there was no information about the exact numbers of patients with traditional risk factors such as hypertension, hyperlipidemia, etc. There was only information that the stroke was cryptogenic. Therefore, according to your suggestion, we added a sentence in the manuscript stating that “In most of the studies the included patients did not have any cardiovascular and cerebrovascular risk factors.

Reviewer 2 Report

Review Report

In this manuscript, the authors elaborate on the role of inherited predisposition to thrombophilia in the etiology of ischemic stroke in young adults using meta-analysis. The authors concluded after statistical analysis that all five genetic markers, Factor V Leiden (FVL), Prothrombin (PT), Protein C (PC), Protein S (PS), Antithrombin (AT), studied were risk factors for ischemic stroke, providing candidate genes for further test, which could improve the quality of care. While the area of research is important, some format and experimental set up/interpretation improvements are recommended.

Major comments:

The authors state in the Abstract and Introduction that this article is about ischemic stroke in young adults, while in the methods section, line 92, the selected literature is screened for the age group 18-65 years. And in Table S1, there are also several papers with a mean age of 61 years old or a range of 53-57 years old, which are in the middle-aged and older age groups. Could the authors please explain whether the criteria for selecting age groups and the definition of "young adults" meet the normative requirements? Otherwise, the data of the above mentioned literatures in Table S1 are not suitable for statistics in this article.

Minor comments:

1. Line #133: “OR=1.74”. It is different from “OR 1.75” in Figure 2. Please modify the text or Figure 2 after double check.

2. Line #153: “The level of heterogeneity was 30%”. It is different from “33%” in Figure 3. Please modify the text or Figure 3 after double check.

3. All forest plots need more figure legends. Such as the meaning of the different shapes (e.g. diamond and square), the meaning of the size of the shape, the meaning of the different colors and the meaning of the horizontal and vertical coordinates.

4. Figure C of Figure S8 is not fully displayed.

5. Formatting issues: Line #177, 185, 186, and 189: Change “I2” to “I2”.

6. The language of the article needs to be improved (especially the paragraph in lines 100-111).

Author Response

Dear reviewer, thank you for your comments. According to recent data, in many countries the age structure has changed in the last couple of decades and we observe a bigger group of people above 50 years old. Also, more and more people above 50 years old have an active lifestyle. Therefore, we included studies about people between 18-65 years old since they represent a major percentage of potential patients. We agree that there are studies with an average age of around 60 years old, but these studies meet the inclusion criteria and there were no patients above 65 years old in any articles, included in the meta-analysis.  

  1. Line #133: “OR=1.74”. It is different from “OR 1.75” in Figure 2. Please modify the text or Figure 2 after double check. - Thank you for your comment. We changed the odds ratio in the text.

  1. Line #153: “The level of heterogeneity was 30%”. It is different from “33%” in Figure 3. Please modify the text or Figure 3 after double check. - Thank you for your comment. We changed the percentage of heterogeneity in the text.

  1. All forest plots need more figure legends. Such as the meaning of the different shapes (e.g. diamond and square), the meaning of the size of the shape, the meaning of the different colors and the meaning of the horizontal and vertical coordinates. - Thank you for your comment. We added more details to the figure legends.

  1. Figure C of Figure S8 is not fully displayed. - Thank you for your comment. Unfortunately, this is a limitation of the RevMan program, used for performing the meta-analysis. If the odds ratio is more than 100, like in this case, it would not appear in the funnel plot, because the horizontal line is up to 100. The reader could check this information from the forest plot about Protein C.

  1. Formatting issues: Line #177, 185, 186, and 189: Change “I2” to “I2”. - Thank you for your comment. We applied superscript formatting to I2.

  1. The language of the article needs to be improved (especially the paragraph in lines 100-111). - Thank you for your comment. We edited the text and corrected all spelling mistakes.

Reviewer 3 Report

The work of Tsalta-Mladenov et.al. demonstrates nicely the importance of thrombophilia, which of the is neglected.
Despite the data from the meta-analysis, it would be good to include hemostaseological aspects. It it unfortunately true that thrombophilia diagnostics do not alter the treatment options in most cases, but a lot of this is caused by a slack of knowledge within the clinical practice. 
The authors could address this more in the discussion and elaborate in the outlook, what studies are a needed to close the gap between a diagnostic evidence and a clinical potential. 

Author Response

  1. Despite the data from the meta-analysis, it would be good to include hemostaseological aspects. – Thank you for your comment. Unfortunately, there was no information regarding the hemostasiological parameters of the patients and we cannot add such data to the present research.
  2. The authors could address this more in the discussion and elaborate in the outlook, what studies are a needed to close the gap between a diagnostic evidence and a clinical potential. – Thank you for comment. We added a new sentence to the Discussion section (lines 210-212).